# Does access to finance condition firms' green investment responses to environmental pressure? Evidence from Vietnam

Dinh Duc Truong*

Faculty of Environmental, Climate Change and Urban Studies National Economics University (NEU), Vietnam

* truongdd@neu.edu.vn

## Abstract

Environmental regulation is widely viewed as a key driver of firms' green investment, yet evidence remains mixed, particularly in emerging economies where financial constraints are pervasive. This study examines whether environmental pressure is associated with firms' green investment and whether this relationship is conditioned by financial constraints. Using firm-level data from the World Bank Enterprise Survey in Vietnam, we estimate nonlinear probability models with interaction terms, controlling for firm characteristics as well as industry and regional fixed effects. The results show that environmental pressure is positively associated with green investment, but the magnitude is modest. Financial constraints significantly reduce the likelihood of green investment and weaken firms' responses to environmental pressure. Marginal effects indicate that regulatory pressure is more likely to translate into investment among firms facing lower financial constraints, while the association becomes negligible for financially constrained firms. These findings highlight a conditional mechanism through which environmental regulation operates. Regulatory pressure alone is insufficient to induce widespread green investment when financial constraints are binding. Instead, its effectiveness depends critically on firms' financial conditions. By providing firm-level evidence from an emerging economy, this study contributes to a more integrated understanding of how regulatory and financial factors jointly shape environmental investment behavior.

## 1. Introduction

Environmental sustainability has become an increasingly important concern for firms as governments strengthen environmental regulations and market stakeholders place greater emphasis on responsible production practices [1]. This trend is particularly evident in emerging economies, where rapid industrialization and export-led growth have contributed to environmental degradation and rising energy demand [2]. In this context, firms are expected to respond to environmental pressure by adopting

**Data availability statement:** The data used in this study are from the World Bank Enterprise Survey (WBES) for Vietnam. The Enterprise Survey data are publicly available and can be accessed from the World Bank Enterprise Surveys website (https://www.enterprisesurveys.org).

**Funding:** This study is funded by the National Economics University, Vietnam.

**Competing interests:** The authors have declared that no competing interests exist.

cleaner production processes and investing in environmentally friendly or energy-efficient technologies [3]. Such investments are often viewed as a key pathway toward reducing environmental impacts while maintaining long-term competitiveness.

In practice, however, firms' responses to environmental pressure vary considerably. While some firms actively invest in green technologies, others exhibit limited or delayed reactions, even when operating under similar regulatory conditions [4,5]. This heterogeneity is especially pronounced in developing and emerging economies, where firms frequently face multiple constraints related to finance, technology, and institutional capacity [6]. As a result, the presence of environmental regulations alone does not guarantee that firms will undertake green investments.

A substantial body of literature has examined the relationship between environmental regulation and firms' environmental behavior. Early studies emphasized the compliance costs of environmental regulation and raised concerns about potential negative effects on firm competitiveness [7]. More recent research, influenced by the Porter hypothesis, suggests that environmental regulation can stimulate innovation and encourage firms to adopt cleaner technologies, thereby improving both environmental and economic performance [8,9]. Despite this shift, empirical findings remain mixed, particularly in the context of developing economies [10]. These mixed results suggest that the effectiveness of environmental pressure depends on firm-specific and contextual factors.

At the same time, another strand of literature highlights the importance of financial conditions in shaping firms' investment decisions. Green investments typically involve substantial upfront costs, longer payback periods, and higher levels of uncertainty compared with conventional investments. Consequently, firms' ability to undertake such investments is closely linked to their access to finance [11,12]. Firms facing credit constraints may be unable to respond to environmental pressure through investment, even when they recognize the potential long-term benefits of cleaner technologies. Empirical studies have shown that financial constraints can significantly hinder firms' innovation activities, including investments related to environmental improvement [13–15].

Despite these insights, much of the existing literature treats environmental regulation and financial constraints as separate determinants of firms' environmental behavior. Relatively few studies explicitly examine how these factors interact, particularly at the firm level in emerging economies. In other words, there is limited empirical evidence on whether financial constraints condition the extent to which environmental pressure translates into green investment. Addressing this gap is important for understanding why firms exposed to similar regulatory environments may exhibit very different investment responses.

Vietnam provides a relevant and informative context for examining this issue. Over the past two decades, Vietnam has experienced rapid economic growth driven by industrial expansion, foreign direct investment, and integration into global value chains. This growth has been accompanied by increasing environmental challenges, including air pollution, energy inefficiency, and rising greenhouse gas emissions [16,17]. In response, the Vietnamese government has strengthened environmental

regulations and promoted cleaner production and energy-efficient technologies as part of its broader sustainable development and green growth strategies [18].

At the same time, access to finance remains uneven among Vietnamese firms. While the financial sector has expanded, many firms, particularly small and medium-sized enterprises, continue to face significant credit constraints [19,20]. These constraints may limit firms' capacity to comply with environmental regulations through investment, even when regulatory pressure is present. Previous studies on Vietnam suggest that financial constraints play an important role in shaping firms' innovation and technology adoption decisions [21,22]. However, empirical evidence linking environmental pressure, financial access, and green investment at the firm level remains limited.

Against this background, this study addresses the following research question: Does access to finance condition the extent to which environmental pressure is associated with firms' green investment? In the empirical analysis, this concept is operationalized as financial constraints. Using cross-sectional firm-level data from the World Bank Enterprise Survey conducted in Vietnam, the study examines whether firms facing lower financial constraints are more likely to respond to environmental pressure by investing in environmentally friendly or energy-efficient technologies.

This study contributes to the literature in several ways. First, rather than relying on aggregate indicators such as emissions, innovation outputs, or ESG scores, it provides firm-level evidence on actual green investment behavior, offering a more direct view of how environmental pressure is translated into concrete investment decisions. Second, while prior research has examined financial constraints in green finance and innovation, this study advances the literature by explicitly modeling financial constraints as a conditioning mechanism that shapes firms' responses to environmental pressure, thereby explaining heterogeneous investment behavior across firms. Third, in contrast to the ESG and sustainability reporting literature, which largely focuses on disclosure and performance metrics, this study emphasizes the investment channel as a core mechanism of environmental response, highlighting how firms operationalize regulatory pressure under varying financial conditions. Finally, by focusing on an emerging economy context, the study provides new evidence on how financial frictions and regulatory pressure jointly influence firm-level green investment, helping explain why similar environmental policies may lead to uneven outcomes across firms.

## 2. Theoretical background, model specification, and hypotheses

### 2.1. Theoretical background

The relationship between environmental pressure and firms' environmental investment has been examined through multiple theoretical lenses [23]. A central argument in the literature is that environmental regulations and related pressures create incentives for firms to adjust production processes and adopt cleaner technologies in order to comply with regulatory requirements and reduce environmental risks [1,2]. From this perspective, environmental pressure acts as an external stimulus that encourages firms to internalize environmental considerations in their investment decisions [24].

However, the strength and effectiveness of this stimulus depend not only on economic incentives but also on the institutional context in which firms operate. Drawing on institutional theory, firms' responses to environmental pressure are shaped by how regulatory signals are perceived, interpreted, and enforced. In particular, regulatory uncertainty, enforcement credibility, and institutional capacity play a critical role in shaping firms' behavioral responses. When regulatory frameworks are stable and enforcement is credible, firms are more likely to internalize environmental expectations and respond through investment. By contrast, weak enforcement or uncertain policy environments may reduce the perceived urgency of compliance and discourage long-term environmental investment. Environmental pressure, therefore, reflects not only formal regulation but also the broader institutional environment that conditions firms' expectations and incentives.

At the same time, firms' responses to environmental pressure depend on their internal capabilities and constraints. While the Porter hypothesis suggests that environmental regulation can induce innovation and improve firm performance under certain conditions [2,8], subsequent studies emphasize that such outcomes are not automatic and may vary across

firms and institutional contexts [9,10]. In developing and emerging economies, firms often face structural constraints that limit their ability to respond proactively to regulatory pressure [6].

Among these constraints, financial constraints play a particularly important role. Green investments, including energy-efficient and environmentally friendly technologies, typically require substantial upfront capital and involve uncertain returns over relatively long time horizons [11,12]. As a result, firms' investment responses to environmental pressure are highly sensitive to their financial constraints. Firms facing lower financial constraints are more likely to absorb compliance costs and undertake green investments, whereas financially constrained firms may rely on short-term compliance strategies or delay investment altogether [25].

This perspective is consistent with the broader literature on firm investment and innovation, which highlights financial constraints as a key determinant of firms' strategic choices [26,27]. Empirical evidence shows that financially constrained firms invest less in innovation and technology upgrading, particularly when investments are risky or intangible in nature [13–15]. Environmental investments share these characteristics, suggesting that financial constraints may weaken the association between environmental pressure and green investment.

Taken together, these insights suggest that environmental pressure and financial constraints should not be viewed as independent determinants of green investment. Rather, financial constraints act as a critical conditioning factor that shapes firms' ability to translate environmental pressure into actual investment behavior. When financial constraints are binding, environmental pressure alone may be insufficient to induce green investment, even when regulatory expectations are clear.

## 2.2. Model specification

Based on the above discussion, this study adopts a conditional framework in which firms' green investment decisions are influenced by environmental pressure, with financial constraints moderating this relationship. Environmental pressure represents the external regulatory and institutional forces that encourage firms to improve their environmental performance. Access to finance reflects firms' ability to mobilize financial resources needed to undertake capital-intensive and risky green investments.

In this framework, environmental pressure is expected to increase the likelihood of green investment, but this effect is conditional on firms' financial capacity. Firms with better access to finance are more likely to respond to environmental pressure by investing in environmentally friendly or energy-efficient technologies, while financially constrained firms may be unable to do so.

To empirically test this framework, the following econometric specification is employed. Let $GI_i$ denotes a binary indicator capturing whether firm i has invested in environmentally friendly or energy-efficient technologies. The baseline model is specified as:

$$GI_i = \alpha + \beta_1 EP_i + \beta_2 FN_i + X_i + \varepsilon_i \tag{1}$$

where $EP_i$ measures the degree of environmental pressure perceived by firm i, $FN_i$ captures the firm's financial constraints, $X_i$ is a vector of firm-level control variables, and $\varepsilon_i$ is an error term.

To examine the moderating role of financial constraints, an interaction term between environmental pressure and financial constraints is introduced:

$$GI_i = \alpha + \beta_1 EP_i + \beta_2 FN_i + \beta_3 (EP_i \times FN_i) + X_i + \varepsilon_i \tag{2}$$

In this specification, the coefficient $\beta_3$ captures whether financial constraints conditions the relationship between environmental pressure and green investment. A positive and statistically significant $\beta_3$ would indicate that environmental pressure is more likely to translate into green investment when firms face fewer financial constraints.

Given the binary nature of the dependent variable, the models are estimated using nonlinear probability models. Consistent with the cross-sectional nature of the data, the analysis focuses on associations rather than causal relationships.

### 2.3. Hypotheses development

**Environmental pressure and green investment.** Environmental pressure, reflected in firms' exposure to environmental regulations and standards, is expected to influence firms' investment behavior by increasing the perceived costs and risks associated with environmentally harmful production practices [4,5,28]. When environmental requirements become more salient, firms may face greater regulatory scrutiny, higher compliance costs, or potential penalties if they fail to meet environmental standards. In this context, investing in environmentally friendly or energy-efficient technologies can represent a strategic response that allows firms to reduce regulatory risks and align their production processes with environmental expectations [6,29].

As discussed in the literature, environmental pressure can encourage firms to adopt cleaner technologies and undertake environmental investment, although the strength of this relationship varies across institutional contexts and firm characteristics. In emerging economies, where regulatory enforcement and firms' capabilities differ widely, environmental pressure may nevertheless function as an important signal that prompts firms to consider green investment as part of their longer-term adjustment strategy [30,31]. Therefore, firms facing higher levels of environmental pressure are expected to be more likely to undertake green investment.

*H1: Environmental pressure is positively associated with firms' green investment.*

**Financial constraints and green investment.** Green investment typically involves substantial upfront capital expenditures, longer payback periods, and higher levels of uncertainty compared with conventional investment projects [7,32]. These features make such investments particularly sensitive to firms' financial conditions. Firms facing financial constraints may lack sufficient internal funds or access to external finance, limiting their ability to mobilize the resources required for environmentally friendly or energy-efficient investments [32,33].

In financially constrained firms, investment decisions are often shaped by short-term liquidity considerations rather than long-term strategic objectives. As a result, even when environmental pressure is present, these firms may postpone or avoid green investment in favor of projects with lower risk or quicker returns [34]. Consistent with this reasoning, financial constraints are expected to reduce the likelihood that firms will engage in green investment.

*H2: Financial constraints are negatively associated with firms' green investment.*

**The moderating role of financial constraints.** While environmental pressure may create incentives for firms to improve their environmental performance, the extent to which these incentives translate into actual green investment is likely to depend on firms' financial constraints [1,3,35]. Firms facing lower financial constraints are better positioned to absorb compliance costs and undertake capital-intensive investments in environmentally friendly or energy-efficient technologies. For these firms, environmental pressure may act as a catalyst that strengthens investment responses and accelerates technological upgrading [36].

In contrast, for firms facing severe financial constraints, environmental pressure alone may not be sufficient to induce green investment [8,13]. Even when regulatory expectations are clear, limited access to finance may prevent firms from responding through investment, leading instead to partial compliance or delayed adjustment [12,28]. This implies that financial constraints condition the relationship between environmental pressure and green investment by weakening firms' ability to translate regulatory pressure into concrete investment action.

*H3: Financial constraints negatively moderate the relationship between environmental pressure and firms' green investment, such that the positive association between environmental pressure and green investment weakens as financial constraints increase.*

Fig 1 summarizes the conceptual framework and hypotheses of the study.

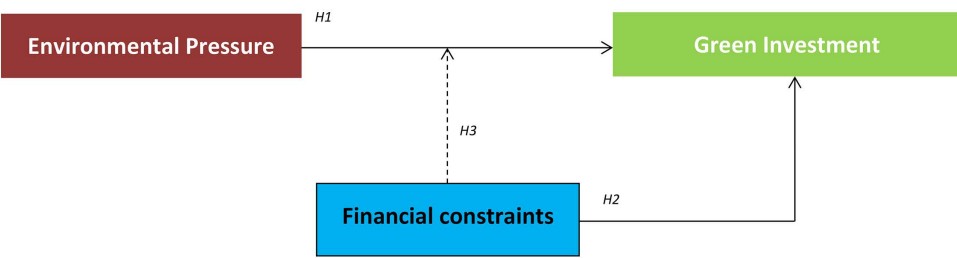

**Fig 1. Conceptual framework.**

## 3. Data, variables, and empirical strategy

### 3.1. Data

This study uses firm-level data from the World Bank Enterprise Survey (WB-ES) conducted in Vietnam in 2023. The Enterprise Survey is a standardized survey program designed to collect detailed information on firms' characteristics, investment behavior, regulatory constraints, and access to finance. It is widely used in empirical research and provides a reliable and comparable basis for analyzing firm behavior in emerging economies.

The WB- ES covers firms operating in both manufacturing and service sectors across major regions of the country. Firms are selected using a stratified random sampling design based on firm size, sector, and geographic location, ensuring that the sample is broadly representative of the non-agricultural private sector [37]. Survey respondents are typically firm owners or senior managers who are expected to have sufficient knowledge of firms' operations, investment activities, and external constraints. The analysis is based on a cross-sectional sample of firms with available information on green investment, environmental pressure, financial constraints, and key firm-level characteristics. Observations with missing information on the main variables are excluded to ensure consistency across model specifications. Given the cross-sectional nature of the data, the study focuses on identifying statistically robust associations rather than causal relationships or dynamic effects over time.

### 3.2. Variable definitions and measurement

**Green investment.** The dependent variable captures firms' engagement in green investment. It is constructed as a binary indicator equal to one if a firm reports having invested in environmentally friendly or energy-efficient technologies during the recent period covered by the survey, and zero otherwise. This measure reflects firms' actual investment behavior rather than stated intentions or attitudes, making it an appropriate proxy for green investment at the firm level. Although this variable does not capture the scale or intensity of investment, it provides a consistent and comparable indicator of whether firms undertake green investment, which aligns with the behavioral focus of the study.

**Environmental pressure.** Environmental pressure is measured using firms' assessments of environmental regulations as an obstacle to their operations. Firms report the extent to which environmental regulations constitute a constraint, ranging from no obstacle to a major or very severe obstacle. This perception-based measure captures the regulatory and institutional pressure experienced by firms in their operating environment.

Using perceived environmental pressure is appropriate in the context of firm-level analysis, as firms' investment decisions are influenced not only by formal regulations but also by how these regulations are perceived and internalized by managers. This approach is consistent with prior studies that emphasize the role of perceived regulatory pressure in shaping firms' environmental and investment behavior.

**Financial constraints.** Financial constraints constitute the key conditioning variable in the analysis. They are measured using firms' responses to the Enterprise Survey question on access to finance as an obstacle to current

operations. Firms evaluate the severity of this obstacle on an ordered scale, with higher values indicating more severe financial constraints.

This measure captures firms' effective financing conditions, reflecting both firm-specific characteristics and broader financial market constraints. Although the survey question is framed in terms of access to finance, the variable is interpreted and used consistently as financial constraints throughout the empirical analysis. Conceptually, access to finance can be understood as the inverse of financial constraints; however, to maintain consistency with the measurement, all results and interpretations are expressed in terms of financial constraints.

**Control variables.** A set of firm-level control variables is included to account for observable heterogeneity that may influence green investment decisions.

Firm size is measured as the logarithm of the number of full-time employees, capturing differences in organizational capacity and resource availability. Firm age, measured as the number of years since establishment, reflects accumulated experience and learning effects. Export status is included as a binary indicator, as firms engaged in export markets may face stronger environmental requirements from foreign buyers. Foreign ownership is captured using a binary variable indicating whether a firm has foreign equity participation, reflecting potential access to advanced technologies and managerial practices.

In addition, industry and regional fixed effects are included in all specifications to control for unobserved sectoral characteristics and local business environment conditions, such as differences in regulatory enforcement, infrastructure quality, and technological opportunities. It is important to note that both environmental pressure and financial constraints are measured using perception-based responses from firm managers. While such measures are commonly used in Enterprise Survey-based research, they may raise concerns regarding common method bias due to correlated reporting behavior. To mitigate this issue, the analysis incorporates a comprehensive set of firm-level controls, as well as industry and regional fixed effects, which help account for systematic differences in reporting across firms. In addition, robustness checks using alternative proxies for financial constraints yield consistent results, reducing the likelihood that the findings are driven by a single measurement approach.

### 3.3. Empirical strategy

The empirical strategy is designed to examine the association between environmental pressure and firms' green investment decisions, with financial constraints acting as a moderating factor. Given that the dependent variable is binary, the analysis employs nonlinear probability models that are appropriate for discrete outcomes.

Logit regression is used as the primary estimation technique. This approach allows for a flexible relationship between explanatory variables and the probability that a firm undertakes green investment. Robust standard errors are applied to account for potential heteroskedasticity. All model specifications include industry and regional fixed effects to mitigate bias arising from unobserved heterogeneity across sectors and locations.

To test the moderating role of financial constraints, interaction terms between environmental pressure and financial constraints are incorporated into the regression models. In nonlinear models such as logit, interaction effects cannot be interpreted directly from coefficient estimates alone. Therefore, the analysis emphasizes marginal effects and predicted probabilities to provide an intuitive interpretation of the conditional relationship. Specifically, marginal effects of environmental pressure are computed at different levels of financial constraints, holding other covariates constant.

Several robustness checks are conducted to assess the reliability of the results. First, alternative proxies for financial constraints available in the Enterprise Survey, such as credit denial or the presence of bank loans, are employed to ensure that the findings are not driven by a specific measure. Second, the sample is split by firm size to examine whether the moderating role of financial constraints differs between smaller and larger firms, given their distinct financing conditions. Third, probit models are estimated as an alternative specification to verify that the main results are not sensitive to the choice of estimation method.

 

Finally, the cross-sectional nature of the data imposes limitations on causal interpretation. While the empirical strategy allows for a careful examination of conditional associations consistent with the conceptual framework, it does not permit causal inference or dynamic analysis over time. Accordingly, the results are interpreted as evidence of statistically robust associations between environmental pressure, financial constraints, and green investment. Accordingly, all results are interpreted as associations rather than causal effects.

## 4. Results

### 4.1. Descriptive statistics

Table 1 presents descriptive statistics for the key variables used in the analysis. Approximately 26.8% of firms report having invested in environmentally friendly or energy-efficient technologies during the reference period. This indicates that green investment remains a selective activity rather than a standard practice among Vietnamese firms, which is consistent with the broader literature on environmental investment in emerging economies.

Firms' perceptions of environmental pressure, measured as the extent to which environmental regulations are considered an obstacle, have a mean value of 2.34 on a five-point scale. This suggests that, on average, firms perceive environmental regulations as a moderate constraint rather than a negligible or overwhelming burden. In contrast, financial constraints exhibit a higher mean value of 2.81, indicating that financial constraints represents a more salient challenge for firms' operations. This difference already points to the potential importance of financial conditions in shaping firms' investment behavior.

Regarding firm characteristics, the average firm employs 81.5 workers, although the distribution is highly skewed, reflecting the coexistence of small enterprises and larger firms in the sample. The average firm age is 13.2 years, suggesting that most firms have accumulated some operational experience. About 23.6% of firms are exporters, and 17.9% report foreign ownership. These figures are consistent with official Enterprise Survey summaries for Vietnam and provide reassurance regarding the representativeness of the sample.

Collectively, the descriptive statistics reveal substantial variation in green investment behavior, perceived environmental pressure, and financial constraints, providing a suitable empirical basis for testing the proposed hypotheses.

### 4.2. Descriptive patterns by green investment status

To provide additional context, Table 2 compares firms that have undertaken green investment with those that have not. Firms engaging in green investment report slightly higher environmental pressure (mean = 2.49) compared with

**Table 1. Descriptive statistics.**

| Variable | Mean | Std. Dev. | Min | Max |
|---|---|---|---|---|
| Green investment | 0.268 | 0.443 | 0 | 1 |
| Environmental pressure | 2.34 | 1.21 | 0 | 4 |
| Financial constraints | 2.81 | 1.27 | 0 | 4 |
| Firm size (log) | 4.40 | 1.32 | 1.61 | 8.01 |
| Firm age (years) | 13.2 | 9.4 | 1 | 62 |
| Exporter | 0.236 | 0.425 | 0 | 1 |
| Foreign ownership | 0.179 | 0.384 | 0 | 1 |

**Note:** This table reports descriptive statistics for the main variables. Green investment is a binary indicator of investment in environmentally friendly or energy-efficient technologies. Environmental pressure and financial constraints reflect firms' assessments of environmental regulations and access to finance as obstacles, with higher values indicating greater severity. All variables are from the World Bank Enterprise Survey (Vietnam).

**Table 2. Descriptive patterns by green investment.**

| Variable | No green investment | Green investment | Difference |
|---|---|---|---|
| Environmental pressure | 2.28 | 2.49 | 0.21** |
| Financial constraints | 2.95 | 2.41 | −0.54*** |
| Firm size (log) | 4.21 | 4.93 | 0.72*** |
| Exporter (%) | 19.8 | 35.4 | 15.6*** |
| Foreign ownership (%) | 15.2 | 26.1 | 10.9** |
| Observations | 1,824 | 668 | |

**Note:** This table compares mean values between firms with and without green investment. Differences indicate mean differences across the two groups. ***, **, and * denote significance at the 1%, 5%, and 10% levels. The table is descriptive and does not imply causality.

non-investing firms (mean = 2.28). Although the difference is modest, it is statistically meaningful and suggests that regulatory pressure may play a role in motivating investment decisions.

More pronounced differences emerge with respect to financial constraints. Firms that have invested in green technologies report significantly lower financial constraints (mean = 2.41) than firms that have not (mean = 2.95). This gap highlights the central role of financing conditions in enabling green investment. Green-investing firms are also larger on average and more likely to export, indicating that scale and international market exposure may facilitate compliance with environmental standards.

These descriptive patterns do not establish causality but provide preliminary evidence consistent with the hypotheses. In particular, they suggest that environmental pressure alone may not be sufficient to induce green investment in the absence of adequate financial capacity.

### 4.3. Baseline regression results

Table 3 reports baseline logit regression results examining the association between environmental pressure, financial constraints, and green investment. Column (1) includes environmental pressure and control variables, while Column (2) adds financial constraints.

Consistent with H1, environmental pressure is positively associated with the likelihood of green investment. In Column (1), the coefficient on environmental pressure is 0.176 and statistically significant at the 5% level. This indicates that firms perceiving stronger environmental regulatory pressure are more likely to invest in environmentally friendly or energy-efficient technologies, holding other factors constant.

To facilitate interpretation, marginal effects are computed and expressed in percentage point terms. The results indicate that a one-unit increase in environmental pressure is associated with an increase of approximately 3–5 percentage points in the probability of green investment, holding other factors constant. This magnitude, while modest in coefficient terms, is economically meaningful given the relatively low baseline rate of green investment among firms.

Financial constraints exhibit a negative and statistically significant association with green investment, supporting H2. In Column (2), the coefficient on financial constraints is –0.271 ($p < 0.01$), suggesting that firms facing more severe financing obstacles are substantially less likely to undertake green investment. The inclusion of financial constraints slightly attenuates the coefficient on environmental pressure, implying that part of the association of regulatory pressure operates through firms' financial constraints

Among control variables, firm size is positively and significantly associated with green investment, while export status also shows a positive effect. Firm age and foreign ownership do not display consistent significance across specifications. The inclusion of industry and regional fixed effects improves model fit, with pseudo $R^2$ values increasing from 0.25 to 0.31.

**Table 3. Baseline logit regressions result.**

| Variables | (1) | (2) |
|---|---|---|
| Environmental pressure | 0.176** | 0.141* |
| Financial constraints | – | −0.271*** |
| Firm size | 0.318*** | 0.291*** |
| Firm age | 0.006 | 0.004 |
| Exporter | 0.412*** | 0.398*** |
| Foreign ownership | 0.087 | 0.072 |
| Industry FE | Yes | Yes |
| Region FE | Yes | Yes |
| Observations | 2,492 | 2,492 |
| Pseudo R² | 0.25 | 0.31 |

**Note:** This table reports logit regression estimates of green investment on environmental pressure, financial constraints, and control variables. All specifications include industry and regional fixed effects. Robust standard errors are used. ***, **, and * denote significance at the 1%, 5%, and 10% levels.

## 4.4. Moderating role of financial constraints

Table 4 presents the core results testing H3, which posits that financial constraints moderate the relationship between environmental pressure and green investment. The interaction term between environmental pressure and financial constraints is negative and statistically significant ($\beta = -0.121$, $p < 0.05$). This indicates that financial constraints weaken the positive association between environmental pressure and green investment. In intuitive terms, this moderating effect means that environmental pressure alone does not automatically lead to green investment. Firms may recognize regulatory requirements or environmental expectations, but without sufficient financial resources, they are less able to translate these pressures into actual investment decisions. By contrast, firms facing lower financial constraints are better positioned to act on environmental signals and undertake the required investments. To facilitate interpretation, marginal effects and predicted probabilities are computed. The results indicate that among firms facing lower financial constraints, an increase in environmental pressure is associated with a higher probability of green investment. In contrast, among firms facing higher financial constraints, the marginal effect of environmental pressure becomes substantially weaker and statistically insignificant.

These findings suggest that environmental regulations are more effective in inducing green investment when firms have lower financial constraints. When financial constraints are binding, regulatory pressure alone appears insufficient to translate into tangible investment behavior. This pattern is fully consistent with the conceptual framework and reinforces the conditional nature of the relationship between environmental pressure and green investment.

## 4.5. Robustness checks

Table 5 reports several robustness checks to assess the stability of the main findings. First, the models are re-estimated using a probit specification. The signs and significance levels remain unchanged. Detailed results for the probit specification are reported in Appendix Table A1.

Second, alternative proxies for financial constraints available in the Enterprise Survey, such as credit rejection and lack of bank loans, are employed. The interaction between environmental pressure and financial constraints remains negative and statistically significant across these specifications, confirming that the moderating effect is not driven by a specific measure.

**Table 4. Moderation effects of financial constraints.**

| Variables | (3) |
|---|---|
| Environmental pressure | 0.284** |
| Financial constraints | −0.198** |
| Environmental pressure × Financial constraints | −0.121** |
| Controls | Yes |
| Industry & region FE | Yes |
| Observations | 2,492 |
| Pseudo R² | 0.21 |

**Note:** This table presents logit regression results including the interaction between environmental pressure and financial constraints. Interaction effects should be interpreted using marginal effects and predicted probabilities. All models include firm-level controls and fixed effects.

**Table 5. Robustness checks.**

| Variables | Probit | Alt. finance proxy | SME subsample |
|---|---|---|---|
| Environmental pressure | + | + | + |
| Financial constraints | − | − | − |
| Interaction | − | − | − |
| Controls | Yes | Yes | Yes |
| Observations | 2,492 | 2,487 | 1,936 |

**Note:** This table reports robustness checks using alternative specifications and samples. The dependent variable is green investment. Results are qualitatively similar across models, supporting the robustness of the main findings.

Third, the sample is split by firm size. The moderating role of financial constraints is more pronounced among small and medium-sized enterprises, whereas the interaction effect is weaker and statistically insignificant among larger firms. This heterogeneity is consistent with the expectation that smaller firms face tighter financing constraints and are therefore more sensitive to financial conditions when responding to environmental pressure. Across all robustness checks, the core results remain qualitatively similar, strengthening confidence in the reliability and internal consistency of the findings.

Taken together, the results provide strong and consistent support for the study's hypotheses. Environmental pressure is positively associated with firms' green investment decisions, but this relationship is conditional on firms' financial constraints. Financial constraints not only reduce the likelihood of green investment directly but also weaken firms' ability to respond to environmental pressure through investment.

These findings underscore the importance of considering financial conditions when evaluating the effectiveness of environmental regulations at the firm level. The results suggest that regulatory pressure is most effective in promoting green investment when firms have adequate access to financial resources, highlighting the complementary roles of environmental and financial policies in emerging economies.

## 5. Discussions

This study examines whether environmental pressure is associated with firms' green investment decisions and whether this relationship is conditioned by financial constraints. Using firm-level data from the World Bank Enterprise Survey in Vietnam, the results show that environmental pressure is positively associated with green investment, but this association is contingent on firms' financial conditions. In particular, financial constraints not only reduce the likelihood of green investment but also weaken firms' responsiveness to environmental pressure.

The positive association between environmental pressure and green investment supports the view that regulatory and institutional forces can motivate firms to adopt environmentally friendly or energy-efficient technologies. This finding is consistent with the Porter hypothesis and related empirical studies, which suggest that environmental regulation can stimulate innovation and environmental investment under certain conditions [1,2,8,34]. However, the magnitude of this association remains moderate. The estimated marginal effects indicate that a one-unit increase in environmental pressure is associated with an increase of approximately 3–5 percentage points in the probability of green investment. This suggests that while regulatory pressure matters, it is not sufficient on its own to induce widespread investment, particularly in emerging economy contexts.

The negative association between financial constraints and green investment underscores the importance of financing conditions in shaping firms' environmental investment behavior. This result aligns with a large body of literature showing that financial constraints limit firms' ability to undertake capital-intensive and risky investments, including innovation and technological upgrading [11,13,15,28,30]. Given that green investment shares many of these characteristics, such as high upfront costs and uncertain returns, it is particularly sensitive to financial constraints. The findings therefore reinforce the argument that financial frictions represent a key barrier to environmental upgrading at the firm level.

The consistently positive effects of firm size and exporter status provide additional insights into the underlying mechanisms of green investment. Larger firms are more likely to engage in green investment, reflecting greater organizational capacity, resource availability, and ability to absorb investment risks. Similarly, exporting firms exhibit a higher likelihood of green investment, likely due to greater exposure to international standards, environmental requirements, and stakeholder scrutiny. These patterns suggest that firm capability and external accountability play an important role alongside regulatory pressure, consistent with arguments in the literature on corporate environmental behavior and sustainability assurance.

The central contribution of this study lies in identifying the moderating role of financial constraints. The results show that financial constraints weaken the positive association between environmental pressure and green investment, indicating that firms facing fewer constraints are better able to translate regulatory pressure into investment decisions. This conditional relationship helps explain the mixed findings in the literature on environmental regulation and firm behavior. While some studies report positive effects of regulation, others find limited or insignificant impacts, particularly in developing countries [6,10,36]. The present findings suggest that such differences may reflect variation in firms' financial constraints rather than inconsistencies in regulatory effects.

The moderating pattern also points to potential nonlinear dynamics. The marginal effects indicate that the association between environmental pressure and green investment becomes substantially weaker at higher levels of financial constraints. This suggests the presence of a threshold beyond which financial constraints become binding, limiting firms' ability to respond to regulatory pressure through investment. Although the current analysis does not formally estimate threshold models, this pattern highlights an important direction for future research on nonlinear firm responses to environmental regulation.

These results can be further interpreted within a broader governance perspective. Environmental behavior is shaped not only by formal regulatory pressure but also by informal governance mechanisms, including social trust, reputational concerns, and external monitoring [38]. Such mechanisms can reinforce corporate accountability and influence how firms respond to environmental expectations. In this context, environmental pressure operates alongside broader accountability structures that shape firms' incentives and constraints.

This governance perspective can be extended by considering the role of sustainability reporting and transparency. In emerging markets, reporting practices often evolve alongside regulatory frameworks and may influence firms' environmental responses. Firms with stronger governance structures and more developed sustainability reporting practices are more likely to internalize environmental expectations and align investment decisions with accountability requirements. Conversely, weaker reporting environments may limit the extent to which regulatory pressure translates into substantive

environmental action. This suggests that financial constraints and regulatory pressure operate within a broader institutional context, where governance quality and reporting practices play a complementary role.

The findings are particularly relevant in the context of emerging economies, where financial constraints are often more severe and unevenly distributed across firms. In such settings, environmental regulation may not translate into meaningful investment outcomes unless firms have sufficient financial capacity. Evidence from Vietnam reflects this pattern, where rapid industrial growth has increased environmental pressure while financial constraints remains uneven, especially among small and medium-sized enterprises [19–22]. The stronger moderating effect observed among smaller firms further supports this interpretation.

Compared with previous studies using Enterprise Survey data, this study contributes by explicitly modeling the interaction between environmental pressure and financial constraints. While earlier work often treats these factors separately, the present analysis shows that their interaction is central to understanding firms' environmental investment behavior. By adopting a conditional framework, the study provides a more nuanced account of how regulatory and financial factors jointly shape green investment decisions.

Importantly, the results should be interpreted in light of the cross-sectional nature of the data. The analysis identifies statistically robust associations rather than causal relationships. Although the findings are consistent across multiple specifications and robustness checks, they do not establish causality. Future research could extend this analysis by employing panel data or quasi-experimental designs to better address potential endogeneity and reverse causality. Longitudinal data would allow for the examination of dynamic investment responses, while exogenous variation in financial conditions or regulatory shocks could provide stronger identification of causal mechanisms. Overall, the findings contribute to a more integrated understanding of environmental investment behavior in emerging economies. They highlight that environmental pressure alone is insufficient to drive green investment in the presence of financial constraints, and that firms' responses are shaped by a combination of regulatory, financial and governance factors.

## 6. Policy implications

The findings of this study offer several policy-relevant insights for promoting firm-level green investment in emerging economies. The results indicate that environmental pressure is associated with a higher likelihood of green investment, with a marginal effect of approximately 3–5 percentage points, but this association weakens substantially at higher levels of financial constraints. This implies that environmental regulation and financial conditions should not be treated as separate policy domains. Instead, their interaction plays a decisive role in shaping firms' environmental investment behavior.

### 6.1. Implications for environmental regulation

The positive association between environmental pressure and green investment suggests that environmental regulations and standards can motivate firms to adopt environmentally friendly or energy-efficient technologies. However, the magnitude of this association, while economically meaningful, remains moderate and is sensitive to financial constraints. This indicates that regulatory pressure alone is unlikely to generate widespread green investment.

First, environmental regulations should be predictable and credible. Given that the estimated effect is modest, firms' responses are likely to depend on the stability of regulatory expectations. Clear and consistent policy signals can strengthen incentives for long-term investment rather than short-term compliance.

Second, enforcement strategies should account for heterogeneity in financial constraints across firms. Uniform regulatory pressure may lead to uneven outcomes when financially constrained firms are less able to respond through investment. Differentiated or phased approaches may therefore improve effectiveness by allowing firms time to adjust.

Third, environmental regulations can be complemented by informational and technical support. Reducing uncertainty around costs and benefits of green investment can help firms translate regulatory pressure into actual investment decisions, particularly when financial constraints are not binding.

## 6.2. Implications for financial policy and financial constraints

The results show that financial constraints both reduce the likelihood of green investment and weaken firms' responses to environmental pressure. This suggests that policies aimed at reducing financial constraints can significantly enhance the effectiveness of environmental regulation.

First, easing general financial constraints is likely to generate indirect environmental benefits. Improving credit availability and reducing financing frictions can increase firms' ability to undertake long-term investments, including those related to environmental upgrading.

Second, targeted financial instruments can play a complementary role. Credit guarantee schemes, concessional loans, and risk-sharing mechanisms can lower the cost and risk of green investment, thereby strengthening the association between environmental pressure and investment outcomes. In practice, such financial support can be operationalized through the development of green finance frameworks that integrate sustainability criteria into financial decision-making. For example, financial institutions may incorporate environmental risk assessments into lending processes, offer dedicated green credit lines, or develop sustainability-linked financial products that incentivize firms' environmental performance. These mechanisms not only reduce financing barriers but also strengthen the alignment between financial incentives and environmental objectives. In emerging markets, where formal green finance systems are still evolving, such approaches can play an important role in building trust, improving transparency, and encouraging firms to engage in environmentally responsible investment.

Third, financial policies should pay particular attention to small and medium-sized enterprises. The results indicate that financial constraints are more binding for smaller firms, limiting their responsiveness to environmental pressure. Targeted support for SMEs can therefore help reduce disparities in green investment across firms.

## 6.3. Integrating environmental and financial policies

A central implication of this study is the need for greater policy integration. Environmental regulation and financial policy are often designed separately, yet the results suggest that their interaction is critical for achieving investment outcomes at the firm level.

Policies that strengthen environmental standards without addressing financial constraints may have limited effects on actual investment behavior. Conversely, reducing financial constraints can amplify firms' responsiveness to environmental pressure. Coordinated policy frameworks that combine regulatory requirements with financial support mechanisms are therefore more likely to generate meaningful investment responses.

In addition, policy evaluation should account for firms' financial constraints when assessing regulatory effectiveness. Ignoring financial conditions may lead to an incomplete assessment of policy outcomes.

## 6.4. Implications for emerging economies

The findings have broader relevance for emerging economies beyond the Vietnamese context. In many developing countries, firms face increasing environmental pressure alongside persistent financial constraints. The results suggest that without reducing financial constraints, environmental regulation may not translate into substantial investment, particularly among smaller firms.

For emerging economies pursuing green growth strategies, this implies that environmental objectives should be aligned with financial system development. Improving financial access and reducing investment risk can enhance firms' ability to respond to environmental challenges. In this sense, green investment should be viewed not only as an environmental objective but also as a function of financial conditions and firm-level capabilities.

In summary, the policy implications emphasize the complementarity between environmental regulation and financial conditions. Environmental pressure is associated with higher green investment, but its effectiveness depends critically on

the extent of financial constraints. Policies that strengthen regulatory signals while reducing financial constraints are more likely to generate sustained green investment at the firm level.

## 7. Conclusions

This study examines whether environmental pressure is associated with firms' green investment and whether this relationship is conditioned by financial constraints. Using firm-level data from the World Bank Enterprise Survey in Vietnam, the results show that environmental pressure is positively associated with green investment, but the magnitude of this association is modest and varies systematically with firms' financial conditions.

Financial constraints emerge as a central factor shaping firms' environmental investment behavior. They not only reduce the likelihood of green investment directly but also weaken the extent to which firms respond to environmental pressure. In other words, environmental pressure is more likely to be associated with investment among firms facing lower financial constraints, while its effect becomes negligible when constraints are binding. This finding provides a clear explanation for the heterogeneous responses to environmental regulation observed in prior studies.

By adopting a conditional framework, this study contributes to the literature by demonstrating that regulatory and financial factors jointly shape firm-level environmental investment decisions. Rather than acting independently, environmental pressure and financial constraints interact to determine whether regulatory signals translate into substantive investment outcomes. This perspective offers a more nuanced understanding of firms' environmental behavior, particularly in emerging economies where financial constraints remain pervasive.

The findings also reveal important heterogeneity across firms. The moderating role of financial constraints is more pronounced among SMEs, which face tighter financing conditions and are therefore less able to translate environmental pressure into green investment. This underscores the importance of firm-level characteristics in shaping the effectiveness of environmental regulation.

Overall, the results suggest that environmental pressure alone is insufficient to drive green investment when financial constraints are binding. A more effective approach to promoting green investment requires recognizing the complementary roles of regulatory pressure and financial conditions. Accounting for this interaction is essential for advancing sustainable industrial development in emerging economies.

## 8. Limitations and future research

This study provides firm-level evidence on the conditional relationship between environmental pressure, financial constraints, and green investment, but some limitations should be noted.

First, the analysis is based on cross-sectional data, which allows for identifying statistically robust associations rather than causal relationships. The observed patterns may partly reflect unobserved firm characteristics or contextual factors. Future research could employ panel data or quasi-experimental designs, such as regulatory changes or exogenous variation in financial conditions, to better address endogeneity and examine dynamic investment responses over time.

Second, key variables, including environmental pressure and financial constraints, are based on firms' self-reported perceptions. While such measures are widely used and capture how firms interpret regulatory and financial conditions, they may introduce reporting bias or common method bias. Although the inclusion of control variables and fixed effects helps mitigate these concerns, future studies could combine survey-based measures with objective indicators of regulatory enforcement or firm-level financial data to strengthen measurement validity.

Third, the measure of green investment captures whether firms invest in environmentally friendly or energy-efficient technologies, but does not reflect the scale or intensity of such investments. Future research could use more granular indicators, such as investment amounts, technology types, or environmental performance outcomes, to better capture heterogeneity in investment behavior.

Finally, the analysis focuses on a single-country context. While Vietnam provides a relevant setting for examining firm behavior in an emerging economy, the findings may not fully generalize to other institutional environments. Comparative studies across countries, as well as extensions incorporating firm-level capabilities or governance factors, could provide a more comprehensive understanding of the conditions under which environmental pressure translates into green investment.

## Supporting information

**S1 Appendix. Robustness check using probit estimation.**
(DOCX)

## Author contributions

**Conceptualization:** Dinh Duc Truong.

**Data curation:** Dinh Duc Truong.

**Formal analysis:** Dinh Duc Truong.

**Investigation:** Dinh Duc Truong.

**Methodology:** Dinh Duc Truong.

**Project administration:** Dinh Duc Truong.

**Software:** Dinh Duc Truong.

**Supervision:** Dinh Duc Truong.

**Visualization:** Dinh Duc Truong.

**Writing – original draft:** Dinh Duc Truong.

**Writing – review & editing:** Dinh Duc Truong.

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
