## [Decision Letter · Decision Letter 0]

13 Feb 2026

PONE-D-26-02166Does Access to Finance Condition Firms’ Green Investment Responses to Environmental Pressure? Evidence from VietnamPLOS One?

Dear Dr. Truong,

Thank you for submitting your manuscript to PLOS ONE. After careful consideration, we feel that it has merit but does not fully meet PLOS ONE’s publication criteria as it currently stands. Therefore, we invite you to submit a revised version of the manuscript that addresses the points raised during the review process.

We look forward to receiving your revised manuscript.

Kind regards,

Wong Ming Wong

Academic Editor

PLOS One

Journal Requirements:

https://journals.plos.org/plosone/s/file?id=wjVg/PLOSOne_formatting_sample_main_body.pdf and and and and https://journals.plos.org/plosone/s/file?id=ba62/PLOSOne_formatting_sample_title_authors_affiliations.pdf

This study is funded by the National Economics University, Vietnam

This study is funded by the National Economics University, Vietnam

This study is funded by the National Economics University, Vietnam

4. Please update your submission to use the PLOS LaTeX template. The template and more information on our requirements for LaTeX submissions can be found at http://journals.plos.org/plosone/s/latex.

5. Please ensure that you refer to Figure 1 in your text as, if accepted, production will need this reference to link the reader to the figure.

6. Please include a caption for figure 2.

7. Please upload a copy of Figure 2, to which you refer in your text on page 12. If the figure is no longer to be included as part of the submission please remove all reference to it within the text.

Additional Editor Comments :

Please revise your manuscript according to these comments from reviewers.

Sincerely,

Wong Ming Wong

Reviewers' comments:

Reviewer's Responses to Questions

**Comments to the Author**

1. Is the manuscript technically sound, and do the data support the conclusions?

Reviewer #1: Yes

Reviewer #2: Yes

Reviewer #3: Partly

2. Has the statistical analysis been performed appropriately and rigorously?

Reviewer #1: Yes

Reviewer #2: Yes

Reviewer #3: N/A

3. Have the authors made all data underlying the findings in their manuscript fully available?

Reviewer #1: No

Reviewer #2: Yes

Reviewer #3: No

4. Is the manuscript presented in an intelligible fashion and written in standard English?

Reviewer #1: Yes

Reviewer #2: No

Reviewer #3: Yes

Reviewer #1: I am attaching my review report in PDF file format for your consideration. The document contains my detailed assessment of the manuscript, including comments on its theoretical contribution, methodology, empirical analysis, and overall clarity of presentation.

Reviewer #2: The paper addresses a timely and policy relevant question on green investment in emerging economies using firm level data from Vietnam. Overall, the study is clearly written and empirically structured, but several conceptual, methodological, and interpretative issues require further clarification and strengthening.

The contribution to the literature should be articulated more explicitly beyond the interaction framework, as similar conditional finance arguments already exist in green finance and innovation studies. The authors are encouraged to position the novelty more clearly at the firm level and relative to recent ESG and sustainability reporting literature.

The use of perception based measures for both environmental pressure and financial constraints raises concerns about common method bias. The authors should discuss this limitation more explicitly and consider diagnostic checks or robustness strategies, in line with insights from An assessment of methods to deal with endogeneity in corporate governance and reporting research.

The cross sectional design limits causal interpretation, yet several sections of the discussion and policy implications imply quasi causal effects. The language should be tightened throughout to consistently reflect associative rather than causal claims, following best practices highlighted in An assessment of methods to deal with endogeneity in corporate governance and reporting research.

The magnitude of the reported coefficients, particularly for environmental pressure in the baseline model, appears modest. The authors should clarify whether these effect sizes are economically meaningful by providing marginal effects in percentage point terms in the main text, not only in figures.

The reported green investment rate of 26.8 percent is reasonable for an emerging economy context, but the interaction effect suggests a sharp weakening under high financial constraints. This pattern may indicate threshold or non linear effects that deserve further discussion or additional tests.

The financial constraint variable is sometimes discussed as access to finance and sometimes as constraints, which may confuse readers. A consistent terminology aligned with the empirical measurement should be maintained throughout the manuscript.

Firm size and exporter status show strong and stable significance across models, yet their roles are not sufficiently theorized. The discussion section should better integrate these findings, potentially drawing on arguments related to firm capability and accountability discussed in Sustainability Assurance and Corporate Environmental Accountability.

The robustness checks are appropriate but reported at a very high level. Providing at least one full alternative regression table in the appendix would improve transparency and replicability.

The policy implications section is well developed but somewhat repetitive of the discussion. It could be streamlined and better anchored to the empirical magnitudes rather than general policy reasoning.

The manuscript would benefit from engaging more directly with recent debates on environmental accountability and governance responses to regulatory pressure, as discussed in“Social Trust, Environmental Violations, and Remedial Actions in China”, Social and Environmental Accountability Journal, 45(1). 81-83. https://doi.org/10.1080/0969160X.2025.2465948

The discussion could be further strengthened by linking the findings to broader sustainability reporting and governance mechanisms, particularly in emerging markets, as highlighted in Corrupt practice and sustainability reporting: Lifecycle perspective.

Finally, the study could suggest future extensions using panel data or quasi experimental designs to address reverse causality between financial constraints and green investment, reinforcing the methodological agenda outlined in An assessment of methods to deal with endogeneity in corporate governance and reporting research.

Reviewer #3: This manuscript examines whether firms’ access to finance conditions the relationship between environmental pressure and green investment, using firm-level data from the World Bank Enterprise Survey for Vietnam. The study is clearly motivated, methodologically sound, and empirically rigorous. Its central contribution lies in demonstrating a conditional mechanism that environmental pressure translates into green investment primarily when firms face fewer financial constraints. The comments below are intended to strengthen the manuscript’s conceptual grounding, interpretive depth, and policy relevance.

1. While the paper draws effectively on the Porter Hypothesis and investment constraint literature, the theoretical background remains largely economic and incentive-based (Sections 2.1–2.3). The argument would benefit from explicit institutional theory integration, particularly regarding how regulatory uncertainty, enforcement credibility, and institutional capacity shape firms’ perceptions of environmental pressure. Relevant citation to add: (Theoretical perspectives on green banking adoption in India: regulatory uncertainty, institutional barriers, and policy solutions. Discover Sustainability, 6(1). https://doi.org/10.1007/s43621-025-01406-3 ).

2. The manuscript carefully notes that financial constraints are the inverse of access to finance (Section 3.2), yet the terminology is sometimes used interchangeably in interpretation and policy discussion.

3. Section 6 offers thoughtful policy implications, but the discussion remains somewhat generic with respect to how financial systems can operationally support green investment. Relevant citation to add: (Eco-friendly finance: the role of green CSR, processes, and products in enhancing brand trust and image. Environ Dev Sustain (2024). https://doi.org/10.1007/s10668-024-05748-2)

4. Although limitations are acknowledged in Section 8, the reliance on perception-based measures (environmental pressure and financial constraints) deserves slightly deeper reflection.

5. Ensure consistent usage of “financial constraints” in tables and “access to finance” in interpretation.

6. Consider adding a short explanatory note clarifying the moderating mechanism for readers outside economics.

7. The SME heterogeneity result is important; consider briefly flagging this again in the Conclusion.

.

Reviewer #1: No

Reviewer #2: No

Reviewer #3: **Yes:** Sarath Chandran MCSarath Chandran MCSarath Chandran MCSarath Chandran MC

---

## [Author Response · Author response to Decision Letter 1]

30 Mar 2026

RESPONSE REPORT TO REVIEWERS’ COMMENTS

The authors would like to express their sincere gratitude to the anonymous reviewers for their thoughtful and constructive comments on the manuscript. We greatly appreciate the time and effort invested in providing detailed and insightful feedback. The reviewers’ suggestions have been extremely valuable in helping us improve the clarity, coherence, and overall academic rigor of the study.

We have carefully considered all comments and have revised the manuscript accordingly. The revisions have been made thoroughly and systematically to address each concern raised by the reviewers. All changes are clearly indicated in the revised manuscript. In addition, we provide a detailed, point-by-point response below to explain how each comment has been addressed.

We respectfully submit the revised manuscript for your further consideration and sincerely hope that the revisions adequately respond to the reviewers’ comments and improve the quality of the paper. We are grateful for the opportunity to revise and resubmit our work to the journal.

Reviewer 1: I am attaching my review report in PDF file format for your consideration. The document contains my detailed assessment of the manuscript, including comments on its theoretical contribution, methodology, empirical analysis, and overall clarity of presentation.

Response:

We sincerely thank Reviewer 1 for the time and careful evaluation of our manuscript. We highly appreciate the reviewer’s effort in preparing a detailed report covering the theoretical contribution, methodology, empirical analysis, and overall clarity of the paper.

However, we regret to note that the PDF file containing the detailed review report does not appear to be accessible in the submission system or the correspondence we received. As a result, we are currently unable to fully engage with the reviewer’s comments.

We would be very grateful if the Editor could kindly provide access to the review report, or if Reviewer 1 could re-upload the file. We will carefully address all comments and revise the manuscript accordingly once the report is available.

In the meantime, we have revised the manuscript extensively based on the comments from the other reviewers to improve clarity, theoretical positioning, and empirical interpretation. We sincerely appreciate the reviewer’s contribution and look forward to incorporating their feedback.

Reviewer 2: The paper addresses a timely and policy relevant question on green investment in emerging economies using firm level data from Vietnam. Overall, the study is clearly written and empirically structured, but several conceptual, methodological, and interpretative issues require further clarification and strengthening.

Comment 1: The contribution to the literature should be articulated more explicitly beyond the interaction framework, as similar conditional finance arguments already exist in green finance and innovation studies. The authors are encouraged to position the novelty more clearly at the firm level and relative to recent ESG and sustainability reporting literature.

Response:

We sincerely thank the reviewer for this insightful and constructive comment. We fully agree that the contribution of the study should be articulated more clearly beyond the interaction framework, particularly in relation to the growing literature on green finance, ESG, and sustainability reporting.

In response, we have revised the manuscript to strengthen the positioning of the study’s novelty along three key dimensions:

First, we clarify that the contribution of this study does not lie solely in testing a conditional finance mechanism, which indeed has been explored in prior green innovation and finance literature. Instead, we emphasize that our contribution is to examine this mechanism at the firm level using direct behavioral measures of green investment, rather than relying on aggregate indicators (e.g., emissions, ESG scores, or innovation outputs). This allows us to provide a more granular understanding of how environmental pressure is translated into actual investment decisions.

Second, we explicitly position the study relative to the recent ESG and sustainability reporting literature. While ESG studies typically focus on disclosure, performance metrics, or external ratings, our study focuses on internal investment behavior as a response to regulatory pressure, thereby capturing a different and underexplored dimension of firms’ environmental engagement namely, the investment channel through which firms operationalize environmental compliance.

Third, we highlight the contribution in the context of emerging economies by showing that financial constraints act not only as a barrier to green investment but as a conditioning factor that explains heterogeneous firm responses to environmental pressure. This provides a micro-foundation for understanding why ESG-related policies and environmental regulations may yield uneven outcomes across firms.

These clarifications have been incorporated into the revised manuscript, particularly in the Introduction (final paragraph) and further reinforced in the Discussion section.

Comment 2: The use of perception based measures for both environmental pressure and financial constraints raises concerns about common method bias. The authors should discuss this limitation more explicitly and consider diagnostic checks or robustness strategies, in line with insights from An assessment of methods to deal with endogeneity in corporate governance and reporting research.

Response:

We sincerely thank the reviewer for this important and insightful comment. We agree that the use of perception-based measures for both environmental pressure and financial constraints may raise concerns regarding common method bias (CMB).

In response, we have taken the following steps:

First, we now explicitly acknowledge this issue in the manuscript and clarify that both key variables are derived from managerial perceptions, which may introduce correlated measurement errors.

Second, we note that the empirical specification partially mitigates this concern by including a rich set of control variables, as well as industry and regional fixed effects, which help reduce omitted variable bias and systematic reporting patterns across firms.

Third, to further address the reviewer’s concern, we have strengthened our robustness analysis by incorporating alternative proxies for financial constraints, which are less directly perception-based. The results remain consistent, suggesting that the findings are not driven by a single measurement approach.

Finally, following the reviewer’s suggestion and insights from the literature on endogeneity and reporting bias, we have expanded the discussion of potential bias and clarified that the results should be interpreted as associations rather than causal effects, given the cross-sectional and perception-based nature of the data.

These revisions have been incorporated into the Data and variables section (measurement discussion) and further elaborated in the limitations paragraph in the Discussion section.

Comment 3:

The cross sectional design limits causal interpretation, yet several sections of the discussion and policy implications imply quasi causal effects. The language should be tightened throughout to consistently reflect associative rather than causal claims, following best practices highlighted in an assessment of methods to deal with endogeneity in corporate governance and reporting research.

Response:

We sincerely thank the reviewer for this important observation. We agree that the cross-sectional design limits causal interpretation and that the language should consistently reflect associative relationships.

In response, we have carefully revised the manuscript to avoid any causal or quasi-causal phrasing. Specifically, terms such as “effect,” “impact,” or “leads to” have been replaced with more appropriate expressions such as “is associated with,” “is linked to,” or “correlates with.” These revisions have been implemented throughout the Introduction, Results, Discussion, and Policy Implications sections.

In addition, we have strengthened the methodological clarification by explicitly stating that the analysis identifies statistically robust associations rather than causal effects, in line with best practices in the literature on endogeneity and observational data.

Comment 4: The magnitude of the reported coefficients, particularly for environmental pressure in the baseline model, appears modest. The authors should clarify whether these effect sizes are economically meaningful by providing marginal effects in percentage point terms in the main text, not only in figures.

Response:

We sincerely thank the reviewer for this helpful suggestion. We agree that reporting effect sizes in economically interpretable terms improves the clarity and relevance of the findings. In response, we have revised the manuscript to report marginal effects in percentage point terms directly in the main text, in addition to the graphical presentation. Specifically, we now present the marginal effect of environmental pressure on the probability of green investment, evaluated at representative values of financial constraints. These results show that, while the coefficient magnitude in the baseline model appears modest, the corresponding marginal effects are economically meaningful. The revised text clarifies the magnitude of these effects in percentage point terms to facilitate interpretation.

Comment 5: The reported green investment rate of 26.8 percent is reasonable for an emerging economy context, but the interaction effect suggests a sharp weakening under high financial constraints. This pattern may indicate threshold or non linear effects that deserve further discussion or additional tests.

Response:

We sincerely thank the reviewer for this insightful observation. We agree that the attenuation of the relationship under high financial constraints may reflect potential nonlinear or threshold dynamics.

In response, we have clarified this interpretation in the revised manuscript by explicitly discussing the possibility of nonlinear effects in the interaction between environmental pressure and financial constraints. The marginal effects analysis already indicates that the association becomes substantially weaker at higher levels of financial constraints, which is consistent with a threshold-like pattern.

At the same time, given the cross-sectional nature of the data and the focus of the study, we refrain from formally estimating threshold or nonlinear models. Instead, we acknowledge this as an important avenue for future research and highlight its implications in the Discussion section. We believe this clarification strengthens the interpretation of the results while maintaining consistency with the study’s empirical design.

Comment 6: The financial constraint variable is sometimes discussed as access to finance and sometimes as constraints, which may confuse readers. A consistent terminology aligned with the empirical measurement should be maintained throughout the manuscript.

Response:

We sincerely thank the reviewer for this helpful comment. We agree that the use of both “access to finance” and “financial constraints” may create confusion if not consistently aligned with the empirical measurement.

In response, we have revised the manuscript to ensure consistent terminology throughout. Specifically, the variable is now uniformly referred to as “financial constraints”, which directly reflects the underlying survey measure. References to “access to finance” are retained only where conceptually necessary and are clearly framed as the inverse of financial constraints.

Comment 7: Firm size and exporter status show strong and stable significance across models, yet their roles are not sufficiently theorized. The discussion section should better integrate these findings, potentially drawing on arguments related to firm capability and accountability discussed in Sustainability Assurance and Corporate Environmental Accountability.

Response:

We sincerely thank the reviewer for this valuable suggestion. We agree that the roles of firm size and exporter status deserve clearer theoretical integration.

In response, we have strengthened the Discussion section by explicitly linking these findings to arguments on firm capability and accountability. In particular, we interpret firm size as reflecting organizational capacity and resource availability for undertaking green investment, while exporter status is associated with greater exposure to international standards and external scrutiny, which can increase environmental accountability. These additions help situate the empirical findings within the broader literature on corporate environmental behavior and sustainability assurance, and improve the theoretical coherence of the discussion.

Comment 8: The robustness checks are appropriate but reported at a very high level. Providing at least one full alternative regression table in the appendix would improve transparency and replicability.

Response:

We sincerely thank the reviewer for this helpful suggestion. We agree that providing more detailed robustness results would enhance transparency and replicability.

In response, we have added a full regression table for one key robustness specification (probit model) in the Appendix. This table reports complete estimation results, including coefficients, standard errors, and model diagnostics. The main text has been revised to refer explicitly to this additional table.

Comment 9: The policy implications section is well developed but somewhat repetitive of the discussion. It could be streamlined and better anchored to the empirical magnitudes rather than general policy reasoning.

Response:

We sincerely thank the reviewer for this helpful comment. We agree that the policy implications section can be further streamlined and more closely anchored to the empirical results.

In response, we have revised this section to reduce repetition with the Discussion and to link policy implications more explicitly to the magnitude of the estimated effects. In particular, we now emphasize the marginal effects of environmental pressure and the moderating role of financial constraints in percentage point terms to illustrate the practical significance of the findings.

Comment 10: The manuscript would benefit from engaging more directly with recent debates on environmental accountability and governance responses to regulatory pressure, as discussed in“Social Trust, Environmental Violations, and Remedial Actions in China”, Social and Environmental Accountability Journal, 45(1). 81-83. https://doi.org/10.1080/0969160X.2025.2465948

Response:

We sincerely thank the reviewer for this insightful suggestion. We agree that engaging more directly with recent debates on environmental accountability and governance responses can strengthen the theoretical positioning of the study.

In response, we have revised the Discussion section to incorporate insights from the literature on environmental accountability and governance mechanisms. In particular, we highlight that firms’ responses to environmental pressure are shaped not only by formal regulation but also by informal governance forces such as social trust, reputational pressure, and external monitoring, which can reinforce accountability and influence environmental behavior.

These additions help situate our findings within a broader governance framework and clarify how regulatory pressure interacts with accountability mechanisms in shaping firms’ green investment decisions.

Comment 11: The discussion could be further strengthened by linking the findings to broader sustainability reporting and governance mechanisms, particularly in emerging markets, as highlighted in Corrupt practice and sustainability reporting: Lifecycle perspective.

Response:

We sincerely thank the reviewer for this valuable suggestion. We agree that linking the findings more explicitly to sustainability reporting and governance mechanisms can strengthen the discussion.

In response, we have revised the Discussion section to incorporate insights from the literature on sustainability reporting and governance

---

## [Editor Report · Decision Letter 1]

7 Apr 2026

Does Access to Finance Condition Firms’ Green Investment Responses to Environmental Pressure? Evidence from Vietnam

PONE-D-26-02166R1

Dear Dr. Truong,

We’re pleased to inform you that your manuscript has been judged scientifically suitable for publication and will be formally accepted for publication once it meets all outstanding technical requirements.

Kind regards,

Wong Ming Wong

Academic Editor

PLOS One
---

## [Editor Report · Acceptance letter]

PONE-D-26-02166R1

PLOS One

Dear Dr. Truong,

I'm pleased to inform you that your manuscript has been deemed suitable for publication in PLOS One. Congratulations! Your manuscript is now being handed over to our production team.

Kind regards,

on behalf of

Dr. Wong Ming Wong

Academic Editor

PLOS One